# Allopregnanolone: Metabolism, Mechanisms of Action, and Its Role in Cancer

**DOI:** 10.3390/ijms24010560

**Published:** 2022-12-29

**Authors:** Carmen J. Zamora-Sánchez, Ignacio Camacho-Arroyo

**Affiliations:** Unidad de Investigación en Reproducción Humana, Instituto Nacional de Perinatología-Facultad de Química, Universidad Nacional Autónoma de México, Ciudad de México 04510, Mexico

**Keywords:** allopregnanolone, pregnanolone, progesterone, neuroactive steroids, cancer, glioblastoma, membrane progesterone receptor (mPR), PXR, GABA_A_ receptor

## Abstract

Allopregnanolone (3α-THP) has been one of the most studied progesterone metabolites for decades. 3α-THP and its synthetic analogs have been evaluated as therapeutic agents for pathologies such as anxiety and depression. Enzymes involved in the metabolism of 3α-THP are expressed in classical and nonclassical steroidogenic tissues. Additionally, due to its chemical structure, 3α-THP presents high affinity and agonist activity for nuclear and membrane receptors of neuroactive steroids and neurotransmitters, such as the Pregnane X Receptor (PXR), membrane progesterone receptors (mPR) and the ionotropic GABA_A_ receptor, among others. 3α-THP has immunomodulator and antiapoptotic properties. It also induces cell proliferation and migration, all of which are critical processes involved in cancer progression. Recently the study of 3α-THP has indicated that low physiological concentrations of this metabolite induce the progression of several types of cancer, such as breast, ovarian, and glioblastoma, while high concentrations inhibit it. In this review, we explore current knowledge on the metabolism and mechanisms of action of 3α-THP in normal and tumor cells.

## 1. Introduction

Allopregnanolone (3α-THP) is a 5α-reduced metabolite of the steroid hormone progesterone (P4), which was the first hormone characterized in the corpus luteum to maintain pregnancy in mammals [1]. The P4 metabolite 3α-THP and its isomer pregnanolone were isolated from the urine of pregnant women in 1934 [2]. Later, a correlation between the chemical structure and sedative effects of 3α-THP and other steroids was determined [3]. Since then, the synthesis of P4 metabolites, the consequences of their impairing synthesis, and their widely diverse mechanisms of action have been described as a never-ending story in both physiological and pathological conditions [4].

The 5α-reduced P4 metabolites were first described as central regulators of female reproductive function, gestation maintenance, and lactation [5,6,7]. However, other relevant actions of these metabolites, particularly 3α-THP, have been described in females and males. 3α-THP has anti-inflammatory effects [8,9,10] and, in the central nervous system (CNS), it induces cell proliferation and migration of neural and glial cells [11,12] and promotes neurodevelopment in different vertebrates like rodents and sheep [13]. The impairment of 3α-THP synthesis in the CNS has been associated with pathologies such as Parkinson’s and Alzheimer’s diseases, anxiety, and depression [14]. Significantly, such effects are mediated through different mechanisms of action from those of P4.

The pioneering work of Wiebe and coworkers in 2000 indicated that levels of 5α-reduced metabolites of P4 are increased in breast cancer [15,16]. Moreover, such steroids promote tumor progression through different mechanisms of action [17]. Along with this, knowledge of 3α-THP’s effects on neuroprotection and as a proliferative agent in the CNS leads to its study in cancer pathophysiology. In this review, we will focus on the synthesis and mechanisms of action described for the P4 metabolite 3α-THP and summarize evidence of the 3α-THP effects, or their impaired synthesis and mechanisms of action, on the progression of diverse cancer types, particularly glioblastomas. We wrote our literature review according to guidelines proposed by Marco Pautasso in 2013 [18] and the IJMS guidelines.

## 2. Allopregnanolone Metabolism in Normal Tissues

The synthesis of 3α-THP depends on P4 availability in steroidogenic cells. The first rate-limiting step of P4 synthesis is the transport of cholesterol from the endoplasmic reticulum or the cytoplasm to the outer mitochondrial membrane, and then to the inner mitochondrial membrane. In the latter, the P450 side chain cleavage (CYP11A1) catalyzes, as indicated by its name, the cleavage of the C20–C22 side chain from cholesterol to produce pregnenolone and isocaproaldehyde [19]. The mechanism and the proteins involved in the cholesterol transport to the mitochondria are not well defined. Some studies point to a huge complex of proteins that maintain close contact between the membranes of the endoplasmic reticulum and mitochondrial membranes of steroidogenic cells.

Although the mitochondrial cholesterol transport complex differs between steroidogenic tissues, some essential proteins have been identified in tight contact with CYP11A1. Examples of this are the steroidogenic acute regulatory protein (StAR) and the translocator protein of 18 kDa (TSPO) [20,21]. Diverse StAR-related lipid transfer domain-containing proteins have been identified. However, in humans, only two bind sterols: STARD1 and STARD3 [22,23]. In this review, we will focus on STARD1, which is a hydrophobic ~37 kDa protein. Although STARD1 has been broadly detected in the whole mitochondria, studies in the mouse MA-10 tumoral Leydig cell line suggest that STARD1 imports cholesterol only when it is located at the outer mitochondrial membrane [24]. To be functional, STARD1 first enters from the intermembrane space to the mitochondrial matrix to be processed into a ~30 kDa shorter form [25,26]. The structural analysis of such proteins indicates that their N-terminal includes a mitochondrial localization sequence, seconded by the classical α/β grip domain from StAR proteins, and a C-terminal, which comprises the cholesterol-binding pocket [27]. STARD1 is highly hydrophobic and conformationally labile, so deciphering its structural changes for importing cholesterol from the endoplasmic reticulum or cytoplasm to the outer mitochondrial membrane has been difficult. However, its interaction with the oligomerized TSPO channel at the outer mitochondrial membrane and the ATPase family AAA-domain-containing protein 3A (ATAD3A), the link between TSPO, STARD1, and CYP11A1, which is located at the inner mitochondrial membrane, have been described [20].

As a monomer, TSPO is abundantly expressed in steroidogenic cells. It has a five alpha-helix structure with a cholesterol recognition sequence at its C-Terminal. Its location has been detected as a polymer either at the outer or the inner mitochondrial membrane [22,28]. Besides cholesterol transport, levels of TSPO correlate with changes in fatty acid metabolism in steroidogenic cells [29]. Moreover, the inhibition of TSPO directly decreases 3α-THP levels in the Ventral Tegmental Area of female rodents’ brains [30]. ATAD3A also participates in the complex of cholesterol transport. It contains two transmembrane domains (TM): TM1 (a.a. 225–242), involved in its spanning in the outer mitochondrial membrane, and TM2 (a.a. 264–274), for its transmembrane location at the inner mitochondrial membrane, and colocalizes with CYP11A1 [31].

Once cholesterol is transported to the inner mitochondrial membrane, its C20–C22 side chain is cleaved by the CYP11A1. CYP11A1 depends on NADPH as a cofactor to produce pregnenolone and isocaproaldehyde through the catalysis of three subsequent reactions. The first and limiting step in steroid synthesis is the 22-hydroxylation of cholesterol, seconded by its 20-hydroxylation to finally produce pregnenolone as the product of oxidative cleavage [32,33]. Although we focused on cholesterol catabolism, some of its derivatives have also been identified as substrates of CYP11A1 for synthesizing pregnenolone [33]. In addition, in tissues and cells with a barely detected expression of CYP11A1, like brain and glial cells, pregnenolone production is detectable and secreted to the culture medium. In such tissues, pregnenolone production was attributed to the metabolic activity of P450 cytochrome other than CYP11A1 [34].

Then, pregnenolone is isomerized to P4 by the 3β-hydroxysteroid dehydrogenase (3β-HSD), which presents two 3β-HSD isozymes: 3β-HSD1 and 3β-HSD2. They are located in the smooth endoplasmic reticulum and the mitochondria [35]. 3β-HSD has been found at the transmembrane inner mitochondrial membrane and in the intermembrane space, where it could be more active due to its structural configuration being sensitive to the pH conditions [36]. P4 is mainly synthesized in the classical steroidogenic tissues: adrenal glands, testis, ovaries, and placenta [35,37,38], although the presence of P4 metabolism machinery has also been reported in tissues such as lungs, skin, and colon, among others, in humans, rodents and monkeys [39,40,41]. Notably, the 3β-HSD2 isozyme has less affinity for its substrates. In addition, 3β-HSD2 is mainly located in steroidogenic tissues, while 3β-HSD1, which presents high substrate affinity, is mainly expressed in other tissues such as the CNS [19].

The synthesis of 3α-THP from P4 begins with the regulatory step of the reduction of P4 hormone to 5α-Dihydroprogesterone (5α-DHP) by the 5α-reductase enzymes (5α-R), also named 3-oxo-5-alpha-steroid 4-dehydrogenases. In humans, three 5α-R isozymes present homology and are expressed in different tissues (Table 1). However, only 5α-R1 and 5α-R2 have a well-described activity of 5α-reductases, whereas 5α-R3 participates in the N-glycosylation of asparagine residues of membrane proteins [42,43]. In addition, two other proteins have been reported with 5α-reductase activity: the glycoprotein synaptic 2 (GSPN2) and the GSPN2-like. However, less is known about such proteins [44]. Here we will focus on the relevance of 5α-R in 5α-DHP synthesis.

The 5α-Rs have an α-rich structure due to their highly hydrophobic amino acid content. They are embedded in the endoplasmic reticulum [48,49]. 5α-R enzymes catalyze the 5α-reduction of the double bond between C4 and C5 of P4 (Δ4,5-ene position), using NADPH as a cofactor and introducing a hydrogen atom with 5 alpha stereochemistry into the C5 of P4 [48,50]. Besides P4, other steroid hormones, such as testosterone and corticosteroids, are substrates for these isozymes. The two main differences in the biochemical properties of 5α-Rs are their optimum pH for synthesizing 5α-reduced steroids (Table 1) [51] and their affinity for substrates. While 5α-R1 presents a substrate affinity in micromolar ranges, the 5α-R2 presents a significantly higher affinity in a nanomolar range. Additionally, 5α-Rs have a preferred affinity for P4 over testosterone and corticosteroids [52]. According to some authors, such isozymes present tissue specificity, which could explain the relevance of the preferred synthesis of some hormone metabolites over others in specific tissues. For example, the role of 5α-DHP and 3α-THP on the CNS has been broadly investigated. They maintain neural function and inflammation in males and females throughout life, and their synthesis is mainly attributed to 5α-R1 in adulthood [45]. In contrast, the sexual maturation and function of the reproductive system in males are maintained by testosterone and its most potent metabolite in humans, the 5α-dihydrotestosterone, whose synthesis is favored by 5α-R2 [46].

Once the 5α-DHP is synthesized, it is then interconverted to 3α-THP by 3α-hydroxysteroid dehydrogenases (3α-HSD). Isozymes with 3α-HSD activity are members of the Aldo-keto reductases family (AKR), subfamily 1C, which in humans comprises four members (AKR1C1-4) with a protein length of ~37 kDa. They all have a highly conserved structure composed of (α/β)8-barrels with three loops conferring their substrate specificity. They are all coded at the same chromosome by different adjacent genes and share a very high sequence identity [53,54]. The reaction directionality of these enzymes depends on the levels of NAD(P)(H), their cofactor, and the availability of substrates. The AKR1C1-4 isozymes act mainly as reductases because the NAD(P)H is usually higher than the NAD(P)+ in the cells. It has also been demonstrated that NAD(P)H inhibits the oxidative reaction of AKR1C2 [55]. Additionally, it has been reported that AKR1C3 has very little oxidative activity [56]. The reduction of 5α-DHP to 3α-THP is mainly promoted by AKR1C1-2 and AKR1C4, as indicated by kinetic analyses [57]. Figure 1 presents the pathway of the 3α-THP metabolism in most steroidogenic human cells.

AKR1C1-4 isozymes regulate the metabolism of many steroids (androgens and prostaglandins) and xenobiotics. As well as their 3α-HSD activity at the C3 carbonyl group of 5α-DHP, they also promote the reduction or oxidation of other carbonyl groups in the C17 and C20 of the cyclopentanoperhydrophenanthrene structure or at the side chain of steroid substrates. They also have different preferred activities for reducing the other mentioned carbonyl groups (Table 2). Under normal conditions, the AKR1C4 isozyme is the most active, and its expression is restricted to the liver; however, this differs in certain cancers, as will be discussed in the next section [58]. AKR1C1-3, in contrast, presents a wider distribution in classical steroidogenic and nonclassical steroidogenic tissues. In human lymphatic endothelial cells, the synthesis pathway of 3α-THP is favored due to a high expression of the involved enzymes [59].

Once 3α-THP is synthesized, it also serves as a substrate for AKR1C1. 3α-THP has a ketone group at C20, reduced by AKR1C1, the isoenzyme with the most significant activity of 20α-keto reductase. The produced metabolite 5α-Pregnan-3α,20α-diol is much less active than 3α-THP and comprises the first step before conjugation to be excreted [62,63]. Additionally, 3α-THP also serves as a substrate for CYP17A1 to produce androsterone as part of the called “backdoor pathway” to promote the synthesis of the potent androgen 5α-dihydrotestosterone [64].

## 3. Allopregnanolone Metabolism in Cancer

One of the best-described phenomena that differentiate normal from malignant tissues is the Warburg effect, which is also considered one of the classic hallmarks of cancer [65,66]. The Warburg effect is characterized by the enhanced processing of glucose as the principal energy source through glycolysis and lactate production, even in the presence of normal levels of oxygen. In such conditions, normal cells produce pyruvate for further oxidative phosphorylation [67]. It is still under discussion when such a metabolic shift in cancer cells occurs: in the carcinogenesis process, or the progression of cancer. However, overexpression of glycolytic enzymes and glucose transporters has been reported in cancer tissues. In addition, several metabolite regulators of glycolysis, such as fructose 2,6-biphosphate, are overproduced to evade the inhibition mechanisms of glycolysis [68]. It has been described that several changes in the mitochondrial function in cancer cells are essential to this process [69,70].

As mentioned in the first section above, the mitochondria are involved in several limiting steps of steroidogenesis. One of the main changes identified in tumoral mitochondria is the overproduction of reactive oxygen species, which leads to carcinogenesis when they reach the cellular nucleus and cause DNA damage. One important source of reactive oxygen species is the deregulated expression of the respiratory chain complexes [70].

Regarding cholesterol, in the 1950s, cholesterol was studied as a carcinogen [71]. It is now accepted that hypercholesterolemia and a high-cholesterol diet promote cancer development. Additionally, high levels of lipoproteins involved in cholesterol transport are associated with the worst survival rates in gastric cancer, among others [72]. The synthesis de novo of cholesterol is enhanced in the cancer context. Cholesterol is needed as a cellular membrane component in highly proliferative cells and for steroidogenesis. Besides, cholesterol is the precursor of all steroid hormones involved in the progression of endocrine and nonendocrine cancers. This section describes current data about alterations in 3α-THP synthesis during cancer. However, it is essential to add that 3α-THP could also regulate the transcription of several enzymes involved in cholesterol processing and steroidogenesis, as will be reviewed in subsequent sections.

As previously mentioned, a rate-limiting step in the 3α-THP synthesis is cholesterol transport to the mitochondria, along with the expression and correct function of CYP11A1. Regarding CYP11A1, the presence of specific single nucleotide polymorphisms in its gene has been associated with a higher risk of endometrial cancer [73]. Moreover, Fan et al. reported that expression of CYPP11A1 was downregulated in 2754 samples of different types of cancer, such as prostate and colon adenocarcinoma, renal clear cell, hepatocellular, lung squamous cell, and uterine corpus endometrial carcinoma, all of them located at The Cancer Gene Atlas (TCGA) repository [74]. However, even if the level expression of CYP11A1 is low, pregnenolone synthesis could also occur in tumor cells independently of the CYP11A1. It must be considered that the recent work of Christina Lin et al. included both immortalized and glioblastoma cell lines, so that pregnenolone synthesis could be carried out not just in normal but tumor cells as well. Lin and colleagues found the production of pregnenolone in two cell lines (MGM-1 and MGM-3) of glioblastoma, the most common primary malignant brain tumor. In these models, levels of CYP11A1 were low. However, the production of pregnenolone was observed and associated with the activity of a different CYP family protein [34]. In the MA-10 mouse tumoral Leydig cells, the synthesis of pregnenolone and P4 was significantly decreased when ATAD3A was silenced [75]. Together, this evidence indicates that the P4 metabolism is active in different types of cancer, even when the well-characterized enzymes involved in P4 synthesis, such as CYP11A1, have not been detectable. This points to the relevance of a better characterization of the cholesterol transport and its cleavage to pregnenolone, especially in nonclassical steroidogenic tissues. Additionally, in MGM-1 and MGM-3 human glioblastoma cells, 3β-HSD, 5α-R1, 5α-R2, and AKR1C1-3 expression were observed, indicating that besides pregnenolone, other steroids could be synthesized by these cancer cells. Another study in MGM-3 cells reported the production of P4. However, 3α-THP levels were not measured [76].

Significantly, steroidogenesis in the cancer context could be altered in tumor cells but also in the tumor microenvironment components, such as immune cells. In the B16-F10 melanoma and the orthotopic EO771 breast cancer models in mice, Mahata et al. reported that interleukins commonly found in tumor cells and their environment, such as IL-4, promote Cyp11a1 upregulation and an increase in pregnenolone synthesis in tumor-infiltrating immunosuppressive Th2 lymphocytes. The silencing of Cyp11a1 in such cells was correlated with significant tumor growth inhibition [77].

Regarding 3β-HSD, its expression and involvement in many malignancies have been reported. In ovarian cancer, the protective effects of P4 have been reported, and the StAR, CYP11A1, and 3β-HSD expression have been correlated with better patient prognosis [78]. In a breast cancer study with a cohort of 161 patient samples, 3β-HSD1 expression was correlated with breast cancer ER-positive tumors. Interestingly, the expression of 3β-HSD1 was positively associated with a better prognosis and low risk of cancer recurrence. However, such data must be taken cautiously because the cohort is too small [79]. In breast cancer cell lines, positive feedback was reported between the upregulated expression of 3β-HSD1 and the expression of IL-4 in ER-positive and ER-negative breast cancer cells [80]. In patients with hepatocellular cancer, 3β-HSD expression is higher than in the normal liver. Additionally, the expression of 3β-HSD1 and 3β-HSD2 isozymes is present in hepatocellular human cancer cell lines. By functional assays, 3β-HSD silencing diminishes the clonogenicity of such cells [81]. In prostate and testis cancer, its activity is enhanced, and it is even considered as a therapeutic approach in treating prostate and breast cancer [82]. Despite this, in such types of cancer, little is known about the relationships between 3β-HSD expression, the synthesis or levels of 3α-THP in plasma or tumoral tissue, and their pathophysiological relevance. The expression of StAR, CYP11A1 and 3βHSD has also been reported in human endometrial cancer cell lines HHUA (estrogen (ER) and intracellular progesterone receptors (PR) positive; differentiated phenotype) and HOUA-1 (ER and PR negative, undifferentiated phenotype). In both cell lines, the production of pregnenolone and P4 was evident [83]. P4 detection in the cell culture medium appears earlier in HHUA than in HOUA-1 cells. This could be explained by differences in P4 metabolism.

Besides the production of 5α-DHP, the 5α-R isozymes are responsible for synthesizing the potent androgen 5α-dihydrotestosterone. In this sense, the relevance of such isozymes in prostate cancer has been evaluated as a therapeutic target [84,85]. However, 5α-R inhibitor treatment has been associated with the decreased synthesis of 5α-DHP and 3α-THP in the CNS, favoring some cases of depression and the risk of high-grade prostate cancer and decreased libido [86]. Interestingly, it has been reported that P4 promotes the transactivation of the androgen receptor in prostate cancer [87], which cannot discard other P4 metabolites that could activate and exert actions in such types of cancer. The role of 5α-reduced progesterone metabolites in malignancies has been mainly reported in breast cancer by Wiebe and collaborators. These authors reported an augmented synthesis of 5α-DHP and 3α-THP levels in infiltrating ductal breast carcinoma, compared with nontumoral tissue from the same patients [15]. Such data could indicate that 5α-R expression or its activity differs in cancerous versus normal tissue. Comparing the human breast cancer cells MCF-7, T47-D, and MDA-MB-231 with the nontumorigenic breast epithelial MCF-10A cells, the expression of 5α-R and AKR1C1-3 isozymes were higher, and lower respectively in the breast cancer cell lines [88]. Interestingly, subcutaneous administration of 5α-DHP promoted tumor growth in a MDA-MB-231 ER-/PR- breast cancer model implanted into mice [89]. This indicates that P4 metabolites could act through other mechanisms apart from activating PR.

As mentioned above, glioblastoma cells also express 5α-R and AKR1C1-3 isozymes. Their functionality was assumed to be due to the high steroid metabolite levels produced when cells are incubated with cholesterol, testosterone, pregnenolone, or P4. The presence of 3α-THP has also been reported among other steroid metabolites [90,91,92]. In human U87 and U251 glioblastoma-derived cells, 5α-DHP induced cell proliferation and migration through the PR [93]. Additionally, in colorectal cancer biopsies and cell lines, the expression of 5α-R1 is significantly elevated compared with that in normal tissue. Such data correlate the high expression of 5α-R1 with the worst prognosis in patients [94]. The levels of P4 metabolites synthesized by colorectal tissue in humans are yet unknown. However, high 5α-DHP and 3α-THP levels were observed in the adult rat colon. In such tissue, the levels of P4 metabolites were superior to those of testosterone [95]. Figure 2 summarizes the correlation between the overexpression of enzymes involved in 3α-THP synthesis and the prognosis detected in different kinds of cancer patients.

AKR1C1-4 involvement in cancer has been reported, although its association with 5α-DHP to 3α-THP conversion has been poorly studied. AKR1C1 levels are higher in small-cell and other lung cancers in stages I to III than in the adjacent nontumoral tissue. In such contexts, overexpression of AKR1C1 induced an increase in cell viability, migration, invasion, and overexpression of metalloproteases MMP2 and MMP9, involved in extracellular matrix degradation and invasion in the human small-cell lung cancer cell line H446 [96]. In breast cancer samples, the expression levels of AKR1C1-2 were lower when compared with the paired nontumoral tissue of the same patients [16].

Interestingly, AKR1C1 and AKR1C1-2 silencing in the T47-D breast cancer cell line enhanced the effect of P4 on decreasing cell numbers, which indicates that P4 metabolism could affect cellular death or proliferation processes in such cells [97]. AKR1C2 in esophageal squamous cell carcinoma is overexpressed relative to paired normal tissue, and its high expression was related to a reduced survival time. AKR1C2 silencing in the KYSE410 and EC109 esophageal cancer cell lines diminished cell migration and viability, along with decreasing tumor growth, when cells were subcutaneously injected in a nude mice model [98]. In prostate cancer, 3α-THP conversion to 5α-DHT has been hypothesized, but yet, unconfirmed [99]. In the human glioblastoma T98G and U373 cell lines, a temozolomide resistance model—the standard chemotherapeutic agent for such types of cancer—has been developed. AKR1C3 overexpression was observed here and proposed as one of the mechanisms involved in the temozolomide resistance [100]. AKR1C4 was also highly expressed in 58% of a cohort of nasopharyngeal carcinoma, which is associated with a high possibility of relapse [101].Figure 2Synthesis of 3α-THP from pregnenolone and the alteration of the expression of the involved enzymes in different types of cancer. Enzymes participating in each metabolic step are indicated by purple squares. On the left side (blue), the association between the high expression of 3β-HSD, 3α-HSD, and good prognosis of patients with different types of cancer is indicated. In the right side (red space), the different types of cancer in which the high expression of 3β-HSD, 3α-HSD, and 3α-HSD isozymes are related to poor prognosis and cancer progression are presented [16,78,79,81,82,88,89,94,96,100,101].
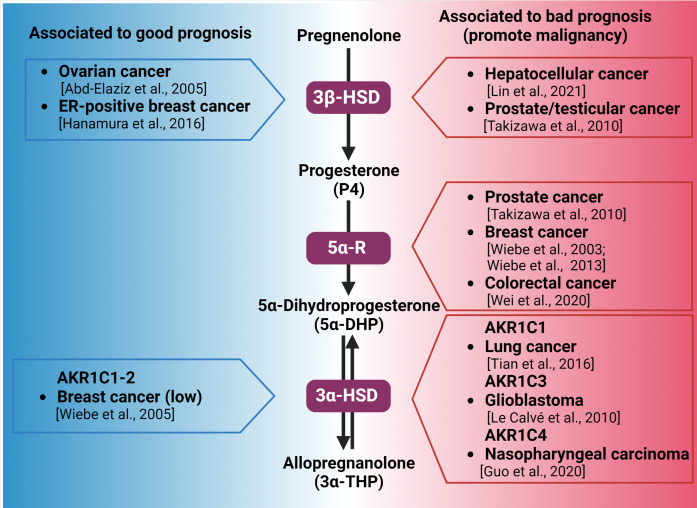


Regarding the study of AKR1C1-4, some considerations must be taken to study the role of such isozymes in the metabolism of cancer. First, some animal models, especially rodents, do not accurately represent AKR1C1-4 activity in humans, because aldo-keto reducase murine isozymes have different preferred substrates and activities [102]. Additionally, some discrepancies have been reported between the expression levels of the AKR1C1-4 mRNA and their protein content [103]. Some other catalytic-independent activities for such enzymes have been described [104].

## 4. Allopregnanolone Mechanisms of Action

The mechanisms of action described for 3α-THP could be grouped into genomic or nongenomic due to the nature of its binding receptor in different cells. It is also important to mention that such effects and mechanisms have mostly been described in the context of neurodevelopment and pathologies of the CNS [105,106].

The PR is a transcription factor of the nuclear receptor superfamily, coded by the PGR gene (also named NR3C3). Two main PR isoforms, coded by such gen under the control of two different promoter sequences, have been reported [107,108]. In addition to its function as a transcription factor, PR interacts with other cytoplasm proteins through the polyproline-rich motifs at their N-terminal domains. An example of this is its interaction with the kinase cSrc [109].

One of the most notable differences between progestogens is their PR affinity, which has been described for P4 (0.35 nM) and 5α-DHP (22 nM). However, binding and gene-reporter assays showed that 3α-THP presents a low direct affinity (>500 nM for the chicken PR, and nondetectable in humans) for PR [110]. Despite this, it has been hypothesized that 3α-THP could regulate the effects of PR due to its interconversion to 5α-DHP in tissues expressing 5α-R and AKR1C1-4 [106,110]. In this section, other reported mechanisms directly activated by 3α-THP will be described.

At the genomic level, 3α-THP has been proposed as a ligand for the Pregnane X Receptor (PXR, also known as Steroid and Xenobiotic Receptor, SXR). PXR is a ligand-activated transcription factor of the nuclear receptors superfamily, coded by the gene NR1I2 [111]. Some variants of PXR have been reported, with different influences on their ligand affinity, dimerization, and transcriptional activity [112]. Although it is constituted by a DNA-binding Domain (DBD) at its N-terminal, and a hinge region, which connects the DBD with the Ligand Binding Domain (LBD) at the C-terminal, its LBD is more flexible than those of other nuclear receptors. Due to its unique pocket binding site, PXR is highly promiscuous to hydrophobic ligands such as steroids. The expression of PXR has been detected by RT-qPCR and RNAseq or microarrays in many tissues, mainly the liver. It has also been detected in the gastrointestinal tract, gonads, uterus, breast, adrenals, bone marrow, smooth muscle, brain, and skin [112,113].

To be functional, PXR forms homodimers and heterodimers, mainly with the retinoic acid receptor (RXR). However, other interactions of PXR with nuclear receptors such as the androgen receptor (AR) have been proposed to promote gene silencing [113,114]. Once activated and in the cellular nuclei, PXR dimers bind to xenobiotic response elements in the promoter or regulator regions of target genes. They recruit coactivators or corepressors for regulating transcription. Many of PXR’s classic targets are involved in cholesterol metabolism and the detoxification of xenobiotics, such as the CYP3A, CYP2B, and CYP20 gene subfamilies [115]. Other targets of PXR are ATP-binding cassette drug transporters [116]. Moreover, it also regulates inflammatory responses and the cell cycle, which are undoubtedly essential processes in the physiopathology of cancer [117]. RNAseq data indicate that PXR is highly expressed in some human cancer cell lines: gastric tubular adenocarcinoma SNU719 cells, colon adenocarcinoma SW403 cells, pancreatic ductal adenocarcinoma ASPC1 cells, and the hepatoblastoma HepG2 cell line [113]. PXR has been implicated in all steps, from carcinogenesis to cancer progress and resistance to therapy [113]. In cancer stem cells from colon cancer, it is overexpressed, and its high levels correlate with the worst survival time and probability of cancer relapse [118]. The expression of PXR was reported in the human ovarian cancer cell lines SKOV-3 and OVCAR-8. When activated, classic PXR targets are overexpressed, along with augmented cell proliferation and tumor weight, in mouse xenografts [119].

The transcription factor function of PXR is activated by 3α-THP, as demonstrated by luciferase activity assays. Interestingly, 3α-THP effects were more significant than those of dehydroepiandrosterone, one androgen metabolite which is a PXR ligand and has similar reported effects to those of 3α-THP in the CNS [112]. This was also observed by Langmade et al. in the cerebellar tissue of the mouse Niemann–Pick C disease npc1−/− model and Chinese hamster ovary cells [120]. It is worth mentioning that Niemann–Pick C disease is characterized by the impairment of cholesterol metabolism and accumulation, in which 3α-THP treatment improved animal survival and reduced inflammatory mediators in a GABA_A_ receptor-independent way [120]. This study clarifies the broad actions of 3α-THP other than the GABA_A_ receptor, which is by far the most characterized mechanism described for 3α-THP, as reviewed below. To our knowledge, there have been no affinity studies performed for 3α-THP to PXR so far.

Moreover, positive feedback between PXR and 3α-THP has been reported. Frye et al. reported that in the Ventral Tegmental Area of female rodents, silencing of PXR with antisense oligonucleotides decreases the levels of 5α-DHP and 3α-THP measured in other brain areas such as the hippocampus [121]. Additionally, in the mice 3xTgAD model of Alzheimer’s disease, 3α-THP significantly increased the expression of PXR and the Liver X receptor (LXR) after 9 months of the establishment of pathology, but decreased it at month 12 [122]. There is no evidence, to date, that 3α-THP could also bind LXR.

3α-THP could rapidly activate or modulate membrane receptors. Although these mechanisms are considered nongenomic, as their primary effects are to modify ion conductance, second messengers, and diverse signaling pathways, they ultimately modify gene expression in a relatively more extended lapse. 3α-THP modulates gamma-aminobutyric acid (GABA) neurotransmission through its GABA_A_ receptor. Notably, neither P4 nor 5α-DHP present affinity for GABA_A_ receptors, as determined by binding assays [123]. The GABA_A_ receptor is a ligan-gated ionotropic channel. A functional GABA_A_ receptor comprises a pentameric channel constituted of five from 19 possible different subunits: α1-6, β1-3, γ1-3, δ, ε, θ, π, and ρ1-3. Most receptors are composed of two α, two β, and a fifth subunit which depends on the function and location of the receptor [124]. The subunit composition of the receptor impacts its functional and pharmacological properties. While the γ subunit is most often located in synaptic GABA_A_ receptors, δ subunits are commonly part of extrasynaptic GABA_A_ receptors, for example, in astrocyte cells. Notably, the presence of δ or γ1 subunits favors 3α-THP affinity for the receptor. However, 3α-THP and other pregnanes bind it allosterically into two putative steroid binding sites in the transmembrane space between the α and β subunits of the receptor [125].

The effects of 3α-THP on such receptors depend on its concentration. At nanomolar concentration (100 nM), 3α-THP is a positive allosteric modulator of the GABA_A_ receptor, while micromolar concentrations (1 to 10 µM) directly activate it [126]. Such effects have immediate consequences in the cells, due to the activation of different signaling pathways and the production of second messengers. The neuroactive steroid tetrahydrodeoxycorticosterone (THDOC), which presents a similar structure and affinity for the GABA_A_ receptor as 3α-THP, promotes the phosphorylation of the common extrasynaptic subunit α4 in its residue S443 depending on the protein kinase PKC activity. Importantly, this favors the membrane localization of the GABA_A_ receptor [125,127].

Once 3α-THP binds to GABA_A_ receptors, the cellular response depends on the expression and activity of effectors, modulators, and cotransporters such as SLC12A2. In mature cells of the CNS, activation or positive modulation of GABA_A_ receptors promotes the influx of Cl- ions, and thus, hyperpolarization, which is an inhibitory signal. This has been the most exploited mechanism of 3α-THP and its pharmacological analogs to induce anxiolytic, anticonvulsive, and sedative effects [123,128]. However, in neural stem cells and pre-progenitor oligodendrocytes, which express cotransporter SLC12A2 that triggers high basal intracellular levels of Cl-, 3α-THP leads to an efflux of Cl- ions when it activates GABA_A_ receptors, and thereby promotes cell depolarization [126]. This effect leads to the activation of voltage-dependent L-type calcium channels, and thus, an increase in the intracellular levels of Ca2+ and cyclic-AMP, promoting the activation of protein kinase a (PKA) and, thus, the activation of transcription factor cyclic AMP-responsive element-binding protein 1 (CREB1) [12,129]. CREB1 regulates the expression of target genes involved in promoting DNA synthesis and cell proliferation [12]. In human Schwann cells, 3α-THP also increases the phosphorylation of CREB, and the expression of glutamate decarboxylase, an enzyme involved in the synthesis of GABA. This is also correlated with an increase in GABA levels [130]. Melfi et al. have reported in the same model that 3α-THP treatment induces a rearrangement of the actin cytoskeleton and cell migration in a GABA_A_ receptor-dependent way and through the activation of Src/p-FAK [11]. Besides this mechanism, 3α-THP also favors the release of gonadotropin-releasing hormone (GnRH) in mouse-immortalized GnRH GT1-1 cells [131]. In physiological conditions, in the rat ovarian nerve plexus–ovary system, these authors also demonstrated that 3α-THP induces the proliferation, angiogenesis, and enhanced activity of the 3-HSD through the GABA_A_ receptor [132,133]. Besides the CNS, GABA_A_ receptors are also expressed in several tissues in physiologic conditions. They are particularly relevant in controlling the liberation of cytokines by immune cells, along with their proliferation and migration [134]. In murine glioma models and human glioblastoma cell lines, GABA_A_ receptors are functional. When treated with its antagonist bicuculine, cell proliferation is highly promoted, compared with cells treated with the agonist muscimol or vehicle conditions [135]. This suggests that GABA_A_ receptors negatively regulate glioblastoma growth.

Importantly, 3α-THP presents a high affinity for membrane P4 receptors (mPRs). The mPRs are membrane proteins of the progestin and adipoQ receptor family (PAQR) with five members: mPRα (PAQR7), mPRβ (PAQR8), mPRγ (PAQR5), mPRδ (PAQR7), and mPRε (PAQR9) with a similar weight of ~40 kDa. Although they are not part of the G protein-coupled receptors family, mPRs present a structure of transmembrane domains [136,137]. To date, modeling and experimentally determining their structure has been complex. However, it is proposed that they are constituted of seven to eight transmembrane domains with a large C-Terminal domain involved in activating G proteins. Moreover, some studies have suggested that some progestogens’ and progestins’ effects mediated by mPRs could also be independent of the activation of G proteins, and they could act more as ligand-activated enzymes with ceramidase activity that produce sphingoid bases as second messengers capable of secondly activating GPCRs [137,138]. Several studies indicate that mPRα, mPRβ, and mPRγ are coupled to inhibitory G-proteins (G_i_) in human cells or olfactory stimulatory G-proteins (G_olf_) in teleost, while mPRδ and mPRε seem to be coupled to stimulatory G-proteins (G_s_) [139,140].

By binding assays, the affinity of 3α-THP for mPRδ (100 nM) and, to a lesser extent, mPRα and mPRβ has been demonstrated (~400 to 500 nM) [140]. 3α-THP is an agonist of mPRδ, mPRα, and mPRβ, and some of its effects depend on mPRs expression levels. The expression of mPRδ and mPRβ is particularly high in several CNS areas such as the hypothalamus, hippocampus, amygdala, and cerebral cortex. In hippocampal neuronal cells, 3α-THP treatment for 15 min increases cAMP levels, which are correlated with the positive modulation of adenylyl cyclase by G_s_ proteins. When triple-negative MDA-MB-231 breast adenocarcinoma cells were transfected with mPRδ and treated with 3α-THP for 20 min, the activation of ERK kinase was observed along with a decrease in apoptosis [140]. In GT1-7 hypothalamic mouse GnRH cells, where mPRα and mPRβ are the most expressed receptors, treatment with 3α-THP in concordance with the activation of G_i_ proteins decreases cAMP levels and cell death [141]. In Figure 3, the most studied mechanisms of 3α-THP to date are presented.

Additionally, crosstalk among the three mechanisms activated by 3α-THP has been proposed. When Melfi et al. reported the effects of 3α-THP on cell migration, a correlation between augmented migration and a dynamic change in the levels of Src and Fak kinases phosphorylation was observed. Interestingly, cotreatment with 3α-THP and the GABA_A_ inhibitor bicuculine promotes higher phosphorylation levels of such kinases than 3α-THP alone or the GABA_A_ agonist muscimol [11]. Additionally, when mPRs were stimulated with the agonist ORG-02-0, migration was promoted in Schwann cells, along with increased activation of Src [142]; this could indicate a complicated activation of several mechanisms involved in the effects of 3α-THP that could compensate each other when one of them is blocked.

Moreover, 3α-THP, but not 5α-DHP or 5α-reduced androgen metabolites, could modulate other neurotransmitter receptors. 3α-THP enhances the activation of dopamine D1 receptors. This was analyzed in a mice model of prepulse inhibition of startle, which is helpful in determining the function of the dopamine receptor. It was reported that D1 receptor agonists, like SKF-82958, impair the prepulse inhibition of startle, and cotreatment with 3α-THP potentiates the effect of SKF-82958. The effect of SKF-82958 was not modified by the pharmacological antagonist of the GABA_A_ receptor bicuculine, or the PXR silencing, suggesting that 3α-THP effects are produced directly through the regulation of the D1 receptor [143]. 3α-THP could also act through glutamate N-methyl-D-aspartate (NMDA) receptors. The release of GnRH and glutamate was induced by a micromolar concentration of 3α-THP in the medium basal hypothalamus and the anterior preoptic area slices of ovariectomized rats. In this study, the NMDA receptor antagonist AP-7 decreased the 3α-THP effect of inducing GnRH and glutamate release, suggesting that the effect of 3α-THP is mediated by its interaction with NMDA receptors [144].

## 5. Effects of Allopregnanolone on Cancer Models

In some in vitro models, the effects of 3α-THP have been determined. In the human U87 glioblastoma-like cells, the metabolite 3α-THP significantly increases the number of cells over 6 days of treatment and promotes proliferation at 72 h of treatment. Interestingly, in such studies, cotreatment with P4 and the 5αR inhibitor, finasteride, almost completely inhibits the effect of P4 [92]. This indicates that besides P4, its metabolites contribute to glioblastoma progression. Using microarrays in the U87 cell line, changes in the gene expression profile under 3α-THP were also assessed. 3α-THP at nanomolar concentration promotes the overexpression of proliferation, DNA reparation, and cytoskeleton rearrangement genes [145]. In addition, in U87 and other glioblastoma cell lines such as U251 and LN229, 3α-THP promotes Src-mediated migration and invasion [146], although the mechanism by which 3α-THP induces Src activation in this model needs to be elucidated. In PC12 rat pheochromocytoma cells, which lack GABA_A_ or NMDA receptors, 3α-THP decreases apoptosis through overexpression of the antiapoptotic proteins Bcl-2 and Bcl-xl. It also induces the activation of PKC [147]. In the breast cancer cell lines MCF-7 and T47D, high concentrations (50 µM) of 3-THP decreased cell viability, while lower concentration (12 µM) stimulated the augmentation of cell numbers, similar to P4 or 5α-DHP [148].

Conversely, in T98G and A172 human glioblastoma cell lines, high micromolar concentrations of 3α-THP alone promote cell death and potentiate the effect of temozolomide, the standard chemotherapeutic agent used for glioblastoma treatment. Such cotreatment also diminished cell migration and invasion through the downregulation of proteins involved in proliferation and integrin signaling [149].

The effects of 3α-THP have been reported in other cancer models. In the human ovarian cancer cell line IGROV-1, 3α-THP promotes proliferation, migration, and clonogenicity at low concentrations [150]. However, the 3α-THP-activated mechanisms in these models are still understudied. Together, the cited in vitro studies analyzed in this review indicate that 1 nM to 20 µM 3α-THP concentrations could induce cancer progression, while higher concentrations inhibit it. Furthermore, changes in the in vivo levels of 3α-THP, and specifically, plasmatic levels of 3α-THP, are not necessarily related to or the cause of cancer. However, the local expression or activity of many enzymes involved in the 3α-THP metabolism is altered in different types of cancer. This could be traduced in an altered synthesis (and levels) of such metabolites in cancer tissues. To state specific concentrations of 3α-THP and their real impact on the tumoral microenvironment, more studies are needed.

In addition to 3α-THP, its synthetic analogs have also been studied in the cancer context. As in the case of 3α-THP, these studies point out that Ganaxolone, a 3-metylated analog, and others promote cell proliferation in different cancer cell lines. Ganaxolone has been proposed as a therapeutic agent for epilepsy because of its high affinity to GABA_A_ receptors, although it also activates mPRs. Ganaxolone has a similar affinity to mPRδ to that reported for 3α-THP (~100 nM) [140,151]. In MDA-MB-231 breast cancer cells (with no expression of GABA_A_ receptors) transfected with mPRδ, Ganaxolone decreased apoptosis and cell death at the same concentration as 3α-THP. In the same model, Ganaxolone also activated the same signaling pathways as 3α-THP: increasing cAMP and activating ERK kinase [151]. Taleb et al. propose the synthesis of 3α-THP analogs with GABA_A_ receptor-agonist effects, but low pro-proliferative action. These authors reported that low concentrations (250 nm to 1 µM) of 3α-THP and its synthetic analogs, BR351 (O-allyl-Allopregnanolone) and BR338 (12-oxo-Allopregnanolone), increase cell viability of the human neuroblastoma cell line SH-SY5Y. The analog BR297 (O-allyl-epi-Allopregnanolone) lacks pro-proliferative activity. All tested steroids, however, present neuroprotective effects and augment GABA conductance [152]. A similar effect for these analogs was observed in primary cultures of mice neural stem cells [153].

To our knowledge, there are no specific reports about sexually dimorphic 3α-THP levels and cancer. However, there are sex differences in cancer prevalence and in neurosteroids, specifically, 3α-THP levels and actions. Although these aspects are not necessarily related, it would be interesting to study them. Regarding many types of cancers, including glioblastomas, their overall incidence is higher in men than women in a proportion of 3:2 (men: women) [154]. Cancer’s aggressiveness, response to treatment, and prognosis are unequal due to the sex effect. It has been reported that genetic sexual differences related to autosomic and sexually specific gene expression influence cancer [155]. However, differences between males and females regarding long-lasting levels of exposure to the endogen sexual hormones and synthetic progestins throughout life in cancer prevalence have been broadly studied. At least in the case of glioblastoma, the most common primary malignant brain tumor, sex hormones impact cancer differently [156,157,158]. While androgens induce glioblastoma progression, the effect of estrogens and progestins depends on their levels, receptor expression, and long-lasting exposure to the hormonal stimulus [156]. Regarding 3α-THP in humans, its plasmatic levels in fertile females vary similarly to P4 and 5α-DHP according to the menstrual cycle phase. The 3α-THP plasmatic levels of females at the follicular phase of the menstrual cycle are similar to those found in healthy adult males (~1 nM). Interestingly, Genazzani et al. evaluated 3α-THP plasmatic levels in males and females of different age ranges (19–39, 40–49, 50–59, and >60 years old). They found that the 3α-THP plasmatic levels of males proportionally decrease as men age. However, female 3α-THP levels at the follicular phase of the menstrual cycle do not vary with age [159].

The evidence presented in this article shows that tumoral cells display very similar responses to those of normal cells when they are exposed to 3α-THP or its analogs at nanomolar to low micromolar concentrations: an increase in cell proliferation, protection against insults, and migration. Although such effects could be beneficial for patients in neurodegenerative states [14,106], cancer cases have to be studied vigilantly [152]. It is worth mentioning that estrogens, and progestins act as inductors of cancer progression and not as carcinogens, at least in glioblastoma, as recently reviewed by Bello-Alvarez et al. [156]. In this line, the P4 metabolite 3α-THP and its analog Ganaxolone display very similar effects to progestins once tumoral growth is established. Moreover, neurological diseases like epilepsy require chronic treatments to improve symptoms and patient quality of life [160]. Under such treatment conditions, the local levels of Ganaloxone could be higher than those needed to induce cancer cell proliferation, although this needs to be investigated.

## 6. Conclusions and Perspectives

State-of-the-art evidence about 3α-THP synthesis and its mechanisms of action in cancer was presented in this review. 3α-THP has been one of the most studied neuroactive steroids with broad actions in neuroprotection, proliferation induction, and immunomodulation. Such characteristics could be exploited by cancer cells. To date, the synthesis of 3α-THP and other P4 metabolites has been confirmed in patient tissue of breast cancer, colon, ovarian, and glioblastoma cell lines, among others. These data suggest an altered synthesis of P4 metabolites in at least some types of cancer, which points to the urgency of generating more data on the P4 metabolism in cancer patients. Moreover, there is little information about the role of other P4 metabolites, 3α-THP, and its analogs, as agents that favor the initiation, promotion, and progression of cancer. This is particularly relevant because analogs of 3α-THP, like Ganaxolone, are approved by the FDA for the long-lasting treatment of epilepsy [160].

To date, few studies about 3α-THP’s role in cancer progression exist. Additionally, 3α-THP presents affinity for a broad class of receptors that could be simultaneously expressed in the same cell or tissue, which makes it difficult to study them. There are plenty of data about 3α-THP effects mediated by its binding to GABA_A_ receptors. However, the crosstalk between GABA_A_ receptors and other mechanisms, like mPRs or PXR activation along with their signaling pathways, needs to be clarified. Additionally, in models where physiological concentrations of P4 induce cancer progression, such as glioblastoma and ovarian cancer, 3α-THP also promotes cellular malignancy, which could reinforce the possible crosstalk between P4, and its 5α-reduced metabolites previously described in breast cancer. Together, these data point to the urgency of illuminating 3α-THP’s effects in cancer pathophysiology by measuring steroid metabolites in cancer patients and studying the potential role of 3α-THP in the progression of cancer malignancy.

## Figures and Tables

**Figure 1 ijms-24-00560-f001:**
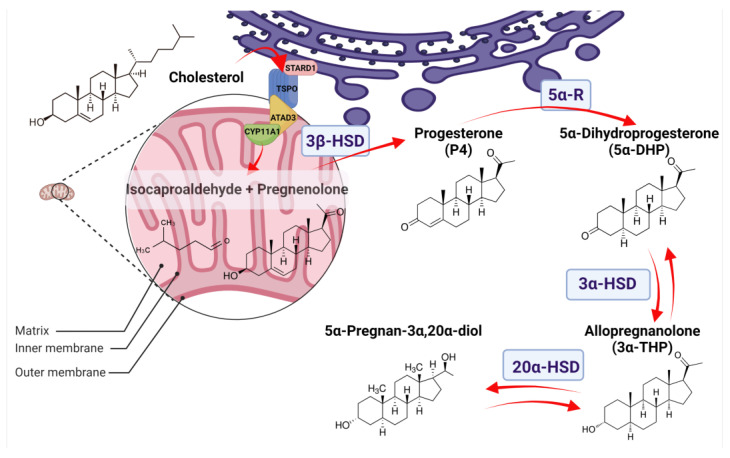
Allopregnanolone (3α-THP) synthesis in normal tissues. The 3α-THP synthesis begins when cholesterol is imported from the cytoplasm or the endoplasmic reticulum (purple) to the mitochondrion (pink) by a protein complex formed by the steroidogenic acute regulatory protein (StARD1), the translocator protein of 18 kDa (TSPO), and the ATPase family AAA-domain-containing protein 3A (ATAD3A), which are in close contact with the P450 side chain cleavage (CYP11A1). CYP11A1, at the inner mitochondrial membrane, catalyzes the cleavage of cholesterol to pregnenolone and isocaproaldehyde. Pregnenolone, either in the inner mitochondrial membrane or at the endoplasmic reticulum, is then isomerized to progesterone (P4) by the 3β-hydroxysteroid dehydrogenase (3β-HSD). At the endoplasmic reticulum, P4 is irreversibly reduced to 5α-Dihydroprogesterone (5α-DHP) by the isozymes 5α-reductases (5α-R). At the cytoplasm, 5α-DHP is reversibly reduced to allopregnanolone (3α-THP) by 3α-hydroxysteroid dehydrogenases (3α-HSD) coded by the AKR1C1-4 genes. Finally, 3α-THP can be a substrate of the 20α-hydroxysteroid dehydrogenase (20α-HSD), coded by AKR1C1, to produce 5α-Pregnan-3α,20α-diol.

**Figure 3 ijms-24-00560-f003:**
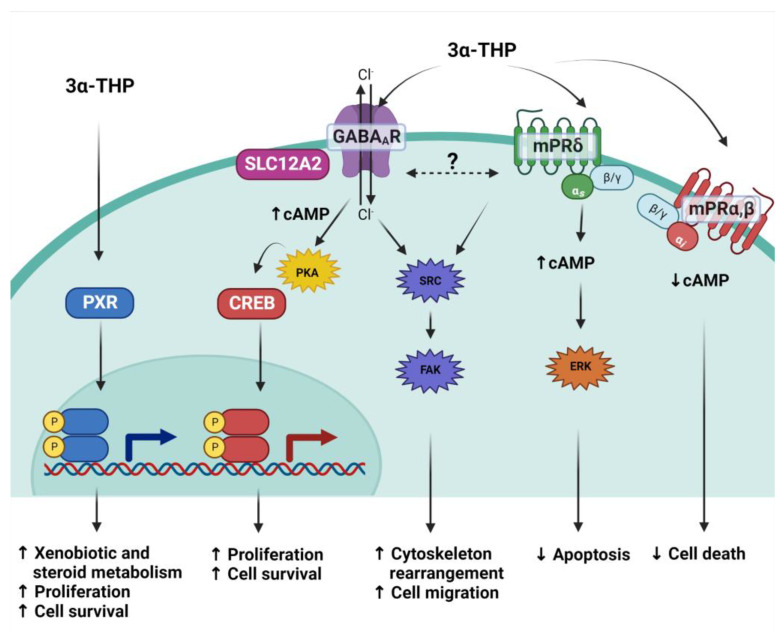
3α-THP mechanisms of action. Due to its lipophilic structure, 3α-THP crosses the cellular membrane and directly activates the Pregnane X receptor (PXR), a transcription factor. It also modulates to the ionotropic pentamer GABA_A_ receptor, which, depending on the expression of the cotransporter SLC12A2, induces changes in the current of chloride ions (Cl^−^), augments cAMP levels and promotes the activation of the PKA kinase and the subsequent activation of the transcription factor CREB. 3α-THP is also a ligand of the membrane P4 receptor mPRδ (green membrane receptor), a transmembrane receptor coupled to G_s_ proteins that augments cAMP. It is hypothesized that both the latter mechanisms interact and regulate the activation of the Src/FAK pathway. Additionally, 3α-THP also binds mPRα and mPRβ (red membrane receptor), which are coupled to G_i_ proteins, and their activation with 3α-THP decreases cAMP levels.

**Table 1 ijms-24-00560-t001:** Comparative characteristics of the 5α-R isozymes involved in the 5α-reduction of steroids.

Isozyme:	5α-R1	5α-R2
Gen/localization	SRD5A1/5p15.31	SRD5A2/2p23.1
Exons number	7	9
Protein weight	29.4 kDa	28.4 kDa
Optimum pH	6–8.5	~5
Human tissue localization	Brain (mainly in adulthood), gastrointestinal tract, liver, and skin.	Almost exclusive in the male reproductive system, liver, and lungs. It is also reported in the brain (mainly in developmental stages: fetal and newborns), and skin.

The data summarized in this table are from the references [45,46,47].

**Table 2 ijms-24-00560-t002:** Comparative characteristics of the AKR1C1-4 isozymes involved in the 3α-reduction of steroids.

Isozyme (Gene Name):	AKR1C1	AKR1C2	AKR1C3	AKR1C4
Gene location (exon number)	10p15.1 (9)	10p15.1 (14)	10p15.1 (10)	10p15.1 (9)
Protein name	20α-(3α)-HSD	3α-HSD type 3	3α-(17β)-HSD type 2	3α-HSD type 1
Preferred activity	1. 3β-keto reductase 2. 20α-keto reductase 3. 3α-keto reductase 4. 17β-keto reductase	3α-keto reductase	1. 3α-keto reductase 2. 17β-keto reductase 3. 20α-keto reductase	3α-keto reductase
Human tissue localization	NS, lungs, liver, mammary glands, testis	NS, lungs, prostate, testis, uterus, mammary glands	Prostate, lungs, liver, prostate, mammary glands, uterus, NS	Liver

The data summarized in this table are from references [58,60,61]. NS: Nervous system.

## Data Availability

Not applicable.

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
