# Peer review of "Allopregnanolone: Metabolism, Mechanisms of Action, and Its Role in Cancer"

_ijms, 2022, doi:10.3390/ijms24010560_

Round 1

Reviewer 1 Report

The manuscript is written clearly with a good workflow. It is comprehensive and contributes well to this interesting and underdeveloped field. This overview lacks information on compliance with the reporting guidelines. In addition, in the "Allopregnanolone metabolism in cancer" section, I recommend adding more information about the possible mechanisms of metabolic changes in malignant growth and focusing on the difference with normal tissues. I recommended accept after minor revision.

Reviewer 2 Report

The authors of the manuscript entitled “Allopregnanolone: metabolism, mechanisms of action, and its role in cancer” summarized the physiological synthesis of allopregnanolone in the organism and its role in cancer.

The review is clearly organized, easy to read and easy to follow. I have not seen such a well and clearly-written review for a long time. It was a pleasure to read it. Thank you!

Despite this, I do have a couple of comments – recommendations for authors for consideration. First, I miss at least a comment on other neurosteroids/analogues of allopregnanolone and their role in cancer. There is no need to extend the review with another chapter, a commentary on literature and reviews would be helpful. Next, the authors did not comment on the topic of gender differences in concentration levels of allopregnanolone and the relevance of this to cancer research. In males and females, neurosteroids are in various concentrations available in the system and as such, one would expect that the biological outcomes might be different. I do not know whether such literature exists and how many, I just miss this topic in this review. Finally, the authors should consider more clear clarification of concentrations of allopregnanolone and its relevance to cancer. The fact that the concentration of allopregnanolone is increased does not necessarily mean that it is related to cancer. Increased concentrations of various neurosteroids and or their metabolites in cancer patients are mentioned. Then, nanomolar and high micromolar concentrations are mentioned throughout the manuscript as the activity concentrations in various in vitro assays. If I understood correctly, the authors suggested that this can be related. However, without specific concentrations mentioned, it is simply a speculative conclusion.

Finally, I would be more careful about conclusions regarding the long-term treatment of epilepsy by Ganaxolone and its relevance of it to allopregnanolone and its effects on the progression of cancer. First, Ganaxolone was already approved by FDA for treatment in 2022, for details see PMID: 35596878. Second, please bear in mind that pediatric patients with CDKL5 deficiency do not respond to any other treatment by regular anticonvulsant drugs. As such, your conclusions suggesting that long-term treatment with Ganaxolone might lead to cancer (overexpressed interpretation) seem to be quite speculative.

Below, please find a list of minor comments, typos, etc.

Line 66:  … in the hole mitochondria …. Misspelled “whole”

Line 128:  … introducing a hydride group in the specific α direction into …. Change to “introducing hydrogen atom with 5 alpha stereochemistry into”

Figure 1. Delete methyl CH3 group naming in the structure of 5a-pregnan-3a,20a-diol and capitalize the letter P

Line 174: cyclopentane-perhydro phenanthrene … no space and misspelt. It is “cyclopentanoperhydrophenanthrene”

Line 186 and 436: …3α THP …. Should be 3α-THP

Line 187: Delete FREE

Line 189: you claim that 3,20-diol is less active than allopregnanolone … in what respect less active?

Line 205: …. Fans and cols …. Change to Fans et al.

Line 211 and 212: …Christina Lin and cols  … Change to Lin et al.

Line 229: …Mahata and cols … Change to Mahata et al.

Line 231 and 232: …. Cyp11a1 … correct to CYP11A1
Line 328: space between the end of the sentence and the reference is missing

Line 333: N terminal …. Should be corrected to “N-terminal”
Line 377: … Langmade and cols … Change to Langmade et al.

Line 386: … Frye and cols …Change to Frye et al.

Line 433 and 487: …Melfi and cols … Change to Melfi et al.

Line 473: …. Levels and ell death …. Change to “cell death”
